# Mechanisms of adipocyte regulation: Insights from *HADHB* gene modulation

**Chaoyun Yang** [ID], **Shuzhe Wang, Yunxia Qi, Yadong Jin, Ran Guan***, **Zengwen Huang** [ID]*

College of Animal Science and Technology, Xichang University, Xichang, China

* guanran@xcc.edu.cn (RG); xndaxue@126.com (ZH)

## Abstract

The *HADHB* gene encodes the beta-subunit of 3-hydroxy acyl-CoA dehydrogenase, closely related to energy metabolism, fatty acid synthesis, and catabolism. This study aimed to investigate the effect of the *HADHB* gene on the proliferation and differentiation of bovine preadipocytes and to gain new insights into the mechanisms of adipocyte regulation. RNA was extracted from adipose tissue of yellow cattle and the HADHB gene CDS region was cloned. Meanwhile, isolation and cultivation of preadipocytes were used for siRNA and adenovirus overexpression, quantitative real-time PCR, transcriptome sequencing, and cell proliferation and cell cycle were measured by oil red staining, CCK8 assay, and flow cytometry. Subsequently, the transcriptome data were analyzed using bioinformatics. The results showed that the *HADHB* gene modulates significantly the expression of critical genes involved in proliferation (*CDK2* and *PCNA*) and differentiation (*PPARγ* and *CEBPα*), influencing preadipocyte proliferation and differentiation and altering cell cycle progression. The results of transcriptome sequencing demonstrated that the overexpression of the HADHB gene markedly altered the transcriptional profile of preadipocytes, with 170 genes exhibiting a significant increase in expression and 113 genes displaying a decrease. The HADHB gene exerts a regulatory influence on the differentiation process of precursor adipocytes by modulating the expression of key genes involved in proliferation and differentiation. These findings highlight the central role of the *HADHB* gene in adipocyte regulation and provide new insights into the regulatory mechanisms governing adipocyte biology.

## 1. Introduction

In mammals, energy homeostasis is primarily coordinated by balancing energy intake and expenditure. Adipose tissue is where excess energy is stored in mammals, and it also participates in thermoregulation, inter-tissue buffering, immune function, inflammation, and pregnancy responses [1–3]. Therefore, regulatory factors involved in adipocyte development have been extensively studied, with peroxisome proliferator-activated receptor gamma (*PPARγ*) receiving the most attention. *PPARγ* is an essential regulator of adipogenesis, responsible for adipocyte differentiation and the maintenance of mature adipocyte morphology [4, 5]. *PPARγ1* and *PPARγ2* can promote adipocyte development [6], but Ren et al. demonstrated that *PPARγ2* plays a decisive role in adipogenic differentiation in 3T3-L1 cells [6]. The *C/*

**Data availability statement:** The data is available from the NCBI database, and its accession

number is PRJNA1192505 and the website is https://www.ncbi.nlm.nih.gov/bioproject/PRJNA1192505.

**Funding:** This study was funded by the Ph.D. Research Launch Project of Xichang University (Grant No. YBZ202211) and Integrative technology and demonstration extension for intensive yak husbandry in Muli County (No. 2023YFN0094).The funders had no role in study design, data collection and analysis, decision to publish, or preparation of the manuscript.

**Competing interests:** The authors declare that they have no competing interests.

**Abbreviations:** CDS, Coding sequence; PBS, Phosphate buffered saline; CG, Oil group; OEG, The *HADHB* overexpression group; TPM, Transcripts per million; TPM, Transcripts per million; DEGs, Differentially expressed genes; FDR, False discovery rate.

*EBP* family of genes has also been extensively studied and is closely associated with adipocyte development, with each member performing distinct roles in the adipocyte differentiation process. For example, *C/EBPβ/δ* is involved in early adipocyte differentiation, while *C/EBPα* primarily plays a role in terminal differentiation [7]. However, the *C/EBP* family of genes is not essential for adipocyte differentiation. In 3T3-L1 preadipocytes, ectopic expression of *C/EBPβ/δ* genes promotes adipogenesis, and overexpression of *PPARγ* can reshape the adipogenic process after *C/EBPα* knockout, but not vice versa [8]. Various molecules involved in adipocyte development have been identified, including FOXO1 [9], FOXp1 [10], SIRT2 [9], miR-188 [11], β3-AR [10], Tm7sf2 [12], G0s2 [13], KLF6 [14], and many others [15–17].

The molecules involved in adipocyte differentiation are not limited to the abovementioned genes. This suggests that searching for new genes that regulate adipocyte development is unnecessary, a concept that led our research group to overlook essential findings from previous studies. We found that the *HADHB* gene was significantly up-regulated in the subcutaneous adipose tissue of high-nutrition-level cattle and was strongly correlated with nutrition level [18]. The *HADHB* gene encodes the β-subunit of 3-hydroxy acyl-coenzyme A dehydrogenase, a key enzyme in β-oxidation [19–21]. So far, research on the *HADHB* gene has mainly focused on diseases. Mutations in the *HADHB* gene can cause mice to have arrhythmias and liver steatosis [22], reduced mitochondrial function and fatty acid oxidation disorders [23], early-onset heart disease and early death [23], and recurrent hypokalemic hypoglycemia [24]. Knocking out the *HADHB* gene can reduce ATP and ROS levels in the central nervous system [25]. In addition, the *HADHB* gene is closely related to fatty acid β-oxidation [21,26–28].

The *HADHB* gene is closely related to energy metabolism, fat synthesis, and decomposition in animals. Studies have shown that the *HADHB* gene is involved in biological processes such as energy metabolism, fat deposition, intramuscular fat, and sugar metabolism in species such as pigs, cattle, sheep, chickens, humans, and mice, and multiple studies have indicated that it may be a critical factor in regulating fat metabolism and muscle function [21,29–31]. At the same time, studies have shown that the expression levels of the *HADHB* gene in buffalo tissues are highest in the liver, followed by adipose tissue and kidney, and are significantly higher than in other tissues [32, 33]. These studies suggest that the *HADHB* gene may play an essential regulatory role in the development of adipocytes. Unfortunately, many of these studies draw conclusions based on enrichment analysis of genomic sequencing results and lack experimental support. At the same time, it has yet to be determined whether the *HADHB* gene regulates the proliferation and differentiation of adipocytes. Therefore, this study aims to investigate the effect of the *HADHB* gene on the proliferation and differentiation of bovine preadipocytes and to use transcriptomic sequencing technology and bioinformatics analysis methods to clarify the changes in the transcriptional profile of preadipocytes induced by overexpression of the *HADHB* gene, providing a new perspective for understanding the regulation mechanism of adipocytes.

## 2. Materials and methods

### 2.1 Ethical approval

The experiments were carried out following the guidelines of the Xichang University animal care committee. All tissues were handled scientifically following harmless disposal methods and principles.

### 2.2 Cloning of the *HADHB* gene CDS region

Subcutaneous fat tissue was collected from cattle, and total RNA was extracted using the Trizol method (Trizol reagent was procured from Sangon Biotech, China.). After extracting

total RNA from the fat tissue, 1.5% agarose gel electrophoresis was used to detect the integrity of the RNA bands, and there was no apparent trailing phenomenon. At the same time, a microplate reader (Bio-RAD680, USA) was used to analyze the $OD_{260}/OD_{280}$ ratio to determine that the $OD_{260}/OD_{280}$ value of all samples was between 1.8 and 2.0. All RNA was reverse transcribed into cDNA using a reverse transcription kit (TaKaRa, Japan) to construct a cDNA library, which was stored at -20°C for later use.

Primer 5.0 and the online tool NCBI blast (https://www.ncbi.nlm.nih.gov/tools/primer-blast/) were used to design primers (F: CCGACGAGACCTAAGGCAGG, R: TCCAGGT-CACCTCTTCTGGAT) based on the cow *HADHB* gene coding sequence (CDS) in GenBank (Accession NO. XM_015473532.2), and the primers were synthesized (Sangon Biotech, China). Subsequently, the *HADHB* gene CDS region was amplified using DNA high-fidelity polymerase (TaKaRa, Japan) with cDNA (2 ng) as the template (pre-denaturation: 95°C for 3 min; 35 cycles ((one step PCR)): denaturation at 98°C for 10 s, annealing at 60°C for 15 s, extension at 68°C for 20 s; final extension at 68°C for 10 min; and cooling at four °C for 10 min), and the amplification product was sequenced for validation (Sangon Biotech, China).

## 2.3 Pre-adipocyte isolation and culture

Pre-adipocytes were isolated using the tissue block method. Adipose tissue was collected from the back of Holstein cattle and washed 3-5 times with phosphate buffered saline (PBS, Hyclone). Under aseptic conditions, the subcutaneous adipose tissue was removed, and the remaining tissue was cut into 1 cm³ pieces and placed in a sterile culture dish. The tissue pieces were then incubated in a humidified incubator at 5% $CO_2$/37°C for 6 hours. After incubation, 6-8 mL of high glucose DMEM (Hyclone) supplemented with 20% FBS serum (BI) and 1% penicillin-streptomycin (Hyclone) was gently added to the tissue blocks. The culture was then maintained in a 5% $CO_2$/37°C incubator for 7-15 days to allow primary pre-adipocyte cells to grow and proliferate.

## 2.4 *HADHB* gene siRNA synthesis and transfection

The siRNA design for the *HADHB* gene (accession number XM_015473532.2) was performed using the siDirect web tool (version 2.0, available at http://sidirect2.rnai.jp/). Three siRNA pairs were designed and labeled siRNA-*HADHB*-1, siRNA-*HADHB*-2, and siRNA-*HADHB*-3. The siRNAs were synthesized by Sangon Biotech (China), and their sequences are shown in Table 1. After the synthesis of siRNA, preadipocytes were transfected when cell confluence reached 30–60%.

## 2.5 Synthesis and transfection of adenovirus Ad-HADHB overexpression

Primers containing KpnI, XhoI, and Kozak sequences (HADHB-KpnI-F: CGGggtaccG-CCACCCCGACGAGACCTAAGGCAGG; HADHB-XhoI-R: CCGctcgagCGGTCCAG-GTCACCTCTTCTGGAT) were designed to amplify CDS region of the *HADHB* gene. The recombinant shuttle vectors, pAdTrack-CMV-*HADHB* and pAd-*HADHB* adenoviral backbone, were constructed by Hanheng Biotech (Guangzhou, China). After obtaining the

**Table 1. siRNA sequences designed for this study.**

|  | sense (5'-3') | anti-sense (5'-3') |
|---|---|---|
| siRNA-*HADHB*-1 | UGGAGAGUGGGUCCGGUCGCAUU | UGCGACCGGACCCACUCUCCAUU |
| siRNA-*HADHB*-2 | GAGUGGGUCCGGUCGCAUCAAUU | UUGAUGCGACCGGACCCACUCUU |
| siRNA-*HADHB*-3 | GAAAUGGAGAGUGGGUCCGGUUU | ACCGGACCCACUCUCCAUUUCUU |

pAd-*HADHB* adenoviral backbone vector, the adenovirus was packaged and amplified by overexpression adenovirus packaging. Once a sufficient adenovirus titer was obtained, a TCID50 (50% tissue culture infectious dose) assay was performed to determine the viral titer before transfection of the preadipocytes.

## 2.6 Cell proliferation and apoptosis assay

**2.6.1 Oil red O staining for adipocyte identification and lipid quantification.** After induction, primary adipocytes were fixed with 10% formaldehyde for 1 hour. Following the removal of formaldehyde, the culture dishes were washed with an equal volume of 60% isopropanol and then stained with Oil Red O working solution (working solution = stock solution: PBS = 4:6) for 30 minutes.

After completion of cell imaging under the fluorescence microscope, the stained wells were washed with 200 μL isopropanol to extract the lipids for analysis. Absorbance values at 480-510 nm wavelengths were measured using a microplate reader to calculate the lipid coefficient.

**2.6.2 Cell viability and proliferation assays.** The Cell Counting Kit-8 (CCK-8) assay (Beyotime, China) was used to measure cell viability. One hundred cells were plated and treated with ten μL CCK-8 reagent, followed by incubation at 37°C with 5% $CO_2$ for one hour. Absorbance was measured at 450 nm to determine cell viability.

The BeyoClick™ EdU-594 assay kit (Beyotime, China) was used to detect proliferating cells. Cells were stained with the EdU reagent by washing with PBS and replacing the medium with freshly prepared DMEM containing ten μM EdU. The cells were then incubated at 37°C with 5% $CO_2$ for two hours. After incubation, the cells were washed with PBS and fixed with 4% paraformaldehyde for 30 minutes at room temperature. DAPI staining was performed for three minutes. After staining, the cells were rewashed with PBS and observed under a microscope.

**2.6.3 Cell cycle assay.** Cell cycle assay was performed using the Beyotime Cell Cycle and Apoptosis Analysis Kit (Beyotime, China). Cells were washed three times with cold PBS solution, and after removing the supernatant, 0.5 mL of 20X PI (propidium iodide) was added to each tube. After sedimentation, the cells were carefully and thoroughly resuspended, and flow cytometry was carried out for analysis.

**2.6.4 Cell apoptosis assay.** The Annexin V-FITC Cell Apoptosis Detection Kit (Beyotime, China) was used for the cell apoptosis assay. According to the manufacturer's protocol, preadipocytes were prepared using the Annexin V-FITC Cell Apoptosis Detection Kit and incubated for 20 minutes at room temperature in the dark. The cell apoptosis rate was then measured using a Biosciences AccuriC6 flow cytometer (BD Biosciences), and the data were analyzed using BD Accuri™ C6 software (version 1.0.264.21; BD Biosciences).

**2.6.5 Western blot.** Adipocytes were collected after 0, 2, and 10 days of induced differentiation. Protein extraction and concentration determination were performed using the BCA protein extraction kit (Solarbio, China). Protein concentrations were adjusted to the same level based on their concentrations. Electrophoresis was performed at 80 V for 90 min, and protein transfer to the membrane was performed at 300 mA for 120 min. The membrane was blocked with 5 g skim milk powder in 100 mL TBST for 90 minutes. The primary antibody (rabbit anti-) was incubated overnight at 4 °C, followed by incubation with the secondary antibody (mouse anti-) at room temperature for 120 minutes. The gel was visualized using a gel imaging system, and images were assessed using Image J and Photoshop software.

## 2.7 Transcriptome sequencing

**2.7.1 Sample collection and sequencing.** After two days of infection with the *HADHB* overexpression group (OEG) (n = 3) and the control group (CG) (n = 3), preadipocytes were

collected, and total RNA was extracted and reverse transcribed into cDNA for transcriptome sequencing using the Illumina HiSeq 4000 platform (Biomarker, China).

**2.7.2 Quality control and alignment.** The quality of the sequencing data was evaluated using FastQC software [34] (version 0.11.7, available at https://www.bioinformatics.babraham.ac.uk/projects/fastqc/). Low-quality, repetitive, and adapter sequences were removed using Trim-galore software (version 0.6.6, available at https://www.bioinformatics.babraham.ac.uk/projects/trim_galore/). Subsequently, clean reads were mapped to the ARS-UCD1.2 reference genome (https://bovinegenome.elsiklab.missouri.edu/downloads/ARS-UCD1.2) using Hisat2 software [35] (version 2.2.1, http://daehwankimlab.github.io/hisat2/). Gene-level quantification was performed using the featureCounts program from the Subread package [36] (version 2.0.1, available at http://subread.sourceforge.net/). The results were expressed as transcripts per million (TPM).

**2.7.3 Differential and functional enrichment analysis.** In this trial, the R package DESeq2 [37] (version 1.24.0, available at http://www.bioconductor.org/packages/release/bioc/html/DESeq2.html) was used to detect differentially expressed genes (DEGs) between the OEG group and the CG group. The fold change (FC) threshold for identifying DEGs was set at 1.5, and the false discovery rate (FDR) was controlled at 0.05.

After identifying DEGs, enrichment analysis was performed. Gene function annotation and visualization were achieved using the R package clusterProfiler [38] (version 4.05). The enrichGO function was selected for Gene Ontology annotation, which includes a biological process (BP), molecular function (MF), and cellular component (CC) categories. The enrichKEGG function was used for the Kyoto Encyclopedia of Genes and Genomes (KEGG) annotation to investigate potential signaling pathways in which the DEGs might be involved. All enrichment analysis results were visualized using the R package ggplot2 [38].

**2.7.4 Identification of essential genes.** Protein-protein interaction analysis of DEGs was performed using the online tool Strings (version 11.0, available at https://string-db.org/) to elucidate the interactions between DEGs. The resulting interactions were visualized using Cytoscape software (version 3.6.1). In addition, essential genes were identified using the Cytoscape plugin CytoHubba, where the intersection of the top 20 DEGs ranked by four methods (Degree, EPC, MCC, MNC) was regarded as the essential genes. In addition, the Cytoscape plugin MCODE was used to identify critical subnetworks and perform functional enrichment analysis, and the resulting seed genes were also categorized as key genes.

## 2.8 Real-time quantitative PCR

The quantitative primers for the *HADHB* (accession number: XM_015473532.2), *CDK2* (accession number: NM_001014934.1), *PCNA* (accession number: NM_001034494.1), *PPARγ* (accession number: NM_181024.2), *CEBPα* (accession number: NM_176784.2) and the housekeeping gene GAPDH (accession number: NM_001034034.2) were designed using Primer 5.0 and the online tool NCBI blast (https://www.ncbi.nlm.nih.gov/tools/primer-blast/), respectively. The primer sequences were shown in S1 Table and synthesized by Sangon Biotech (China).

The quantitative reagent SYBR® Premix Ex Taq II was purchased from TaKaRa (Japan), and the Bio-RAD real-time quantitative PCR instrument (CFX384, USA) was used. The reaction system and program were carried out strictly according to the protocol provided with the reagent kit and the guidelines for quantitative PCR. The real-time quantitative PCR reaction protocol is shown in S2 Table, and the results were calculated using the $2^{-\triangle\triangle Ct}$ method.

## 3. Result

### 3.1 Involvement of the *HADHB* gene in early adipocyte precursor cell differentiation

To investigate the role of the *HADHB* gene in preadipocyte differentiation, preadipocytes were induced to differentiate, and the temporal expression profiles of the differentiation marker genes *FABP4*, *PPARγ,* and the target gene *HADHB* were examined (Fig 1A). *PPARγ* and *FABP4* genes were significantly up-regulated at different time points (2 d, 6 d, 10 d, 12 d) comparing to the undifferentiated state ($P < 0.05$), indicating successful induction of differentiation. However, the expression pattern of the *HADHB* gene showed differences compared to the *PPARγ* and *FABP4* genes. At 2 d and 4 d of differentiation, *HADHB* gene expression was significantly higher than in the undifferentiated state but significantly lower at 4 d compared to 2 d, suggesting the potential involvement of the *HADHB* gene in the early stages of preadipocyte differentiation.

To further elucidate the role of the *HADHB* gene in preadipocyte differentiation, *HADHB* gene knockdown and overexpression experiments were performed. The results (Fig 1B, S1A Fig) showed that knockdown of *HADHB* gene expression significantly inhibited *CEBPα* and *PPARγ* gene and protein levels, regardless of whether preadipocytes were induced to differentiate ($P < 0.05$), with a more substantial inhibitory effect on *PPARγ* gene expression. Furthermore, lipid accumulation experiments showed that *HADHB* gene knockdown resulted in a significant reduction in intracellular triglyceride content and lipid droplet size (Fig 1C). Given the limitations of the siRNA knockdown results, *HADHB* gene overexpression experiments were performed to complement the shortcomings of the knockdown experiments. The results showed that in the OEG group, the expression of *PPARγ* and *CEBPα* genes and proteins was significantly higher than in the CG group after two days of induction ($P < 0.01$), with *PPARγ* showing the highest increase (Fig 1D, S1B Fig). In addition, the OEG group had significantly higher triglyceride (ATG) levels and larger lipid droplets compared to the CG group (Fig 1E). These findings suggest that the HADHB gene may be involved in adipocyte differentiation by regulating the expression of adipogenic marker genes *CEBPα* and *PPARγ*.

### 3.2 Involvement of the *HADHB* gene in the regulation of preadipocyte proliferation

Having confirmed the involvement of the *HADHB* gene in regulating preadipocyte differentiation, we further investigated its effect on cell proliferation. First, the impact of the *HADHB* gene on proliferation-related genes and their protein expression was determined. The results showed that knockdown of the *HADHB* gene significantly decreased the transcription and protein levels of *CDK2* and *PCNA* (Fig 2A, S1C Fig). Conversely, over-expression of the *HADHB* gene significantly down-regulated the transcription and translation of *CDK2* and *PCNA*, with a more pronounced effect on *CDK2* (Fig 2B, S1D Fig). The impact of *HADHB* gene over-expression on preadipocyte proliferation and viability was then assessed using the EdU and CCK-8 assays. The results showed a significant increase in the number of proliferating cells in the OEG compared to the CG (Fig 2C). In addition, the viability of the OEG group was significantly higher than that of the CG at 30h, 42h, and 56h post-transfection (Fig 2D) ($P < 0.05$). These results demonstrate that the *HADHB* gene plays a role in regulating preadipocyte proliferation.

To further elucidate the detailed mechanisms underlying the effect of *HADHB* gene over-expression on preadipocyte proliferation, flow cytometry was performed to assess its influence on the cell cycle. The results (Fig 2E-F) showed that *HADHB* gene over-expression

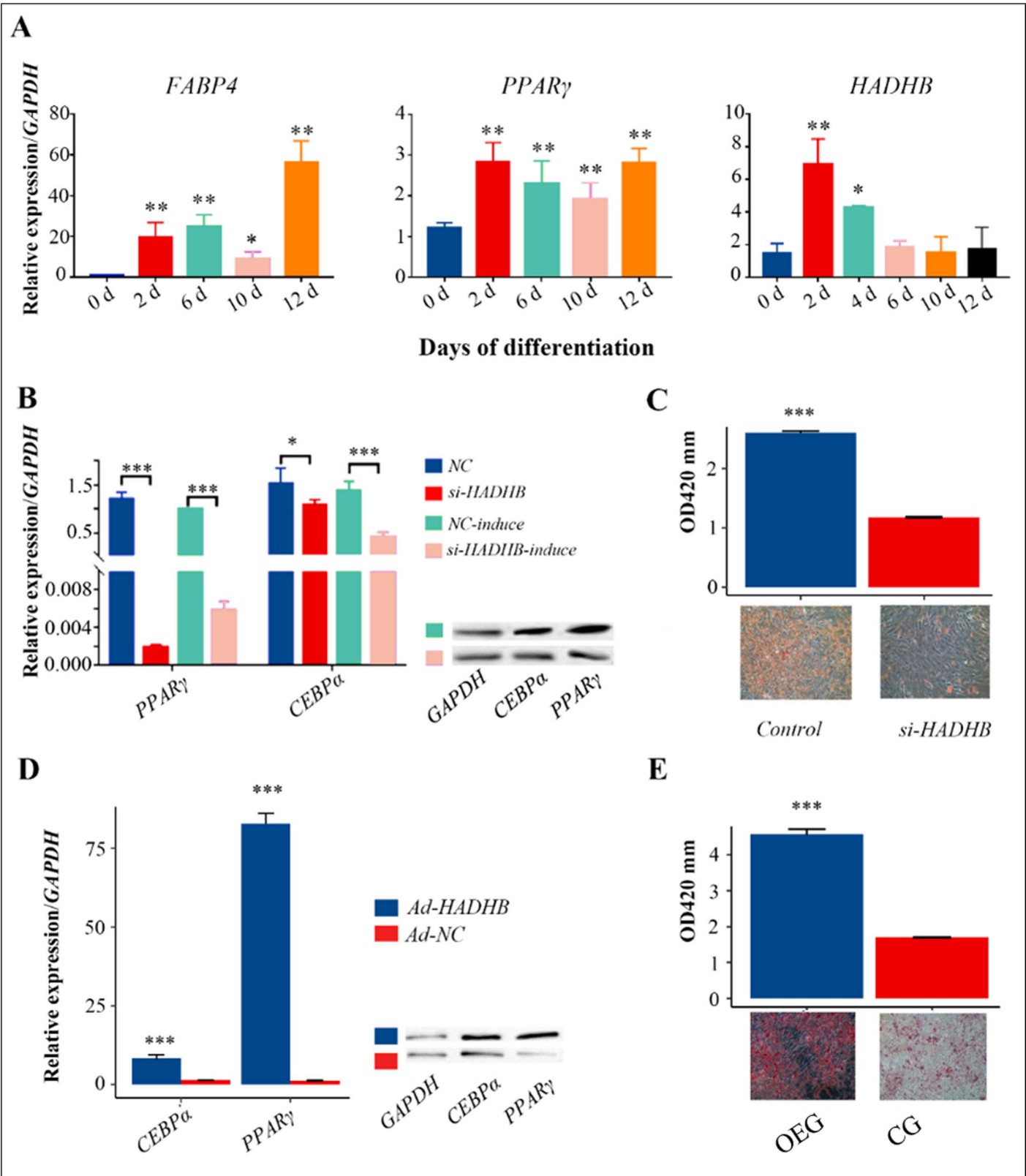

**Fig 1.** A. **Real-time quantitative PCR results of *FABP4, PPARγ,* and *HADHB* gene expression during the induction process of pre-adipocyte differentiation.** B: Effect of si-HADHB on the expression patterns of ***CEBPα*** and ***PPARγ*** genes and proteins in the non-induced and 2-day induced differentiation models. C: Oil

red O staining and ATG quantification of preadipocyte induction and differentiation results for two days with si-HADHB. D: Effects of **HADHB** overexpression on the transcription and protein levels of **CEBPα** and **PPARγ** in pre-adipocytes. E: Effect of **HADHB** over-expression on the adipogenic capacity of pre-adipocytes.

significantly increased the proportion of preadipocytes in the S phase while reducing the G2/M phase (Fig 2G). These observations suggest that the *HADHB* gene primarily involves early to mid-stage preadipocyte proliferation.

### 3.3 Inhibition of *HADHB* gene expression promotes early apoptosis in preadipocytes

Having established that *HADHB* gene knockdown and overexpression can affect preadipocyte proliferation and differentiation, this section further investigates the effect of *HADHB* gene knockdown on preadipocyte apoptosis. The results showed (Fig 3) that inhibition of *HADHB* gene expression resulted in a 2.8% increase in late-stage cell apoptosis (control vs. siRNA, Q2) and a 16.7% increase in early-stage cell apoptosis (control vs. siRNA, Q3). Therefore, inhibition of *HADHB* gene expression primarily induces early-stage apoptosis in preadipocytes.

### 3.4 Transcriptomic changes induced in pre-adipocytes by overexpression of the *HADHB* gene

The sequencing results are presented in S3 Table. Briefly, each sample generated approximately 2 GB of raw data. The average GC content ranged from 51.59% to 52.04%. The quality scores, with a false discovery rate (FDR) below 0.01 (Q20 > 97.98%) and FDR below 0.001 (Q30 > 94.45%), indicated that the library preparation, sequencing, and quality control for this experiment met the requirements for subsequent analysis. The reads were then aligned, and the alignment rate to the reference genome ranged from 93.58% to 95.86%, with a multi-mapping rate between 1.75% and 1.96%. These results confirmed that the reads originated from the bovine genome and demonstrated the reliability of the data. The PCA analysis showed a clear division between the OEG and CG groups (S2 Fig), indicating successful overexpression of the target gene. The apparent clustering of these two groups in the PCA plot confirms the efficacy of the overexpression and its impact on the overall gene expression profile.

Further statistical analysis of these reads revealed that the proportion originating from exons ranged from 80.82% to 81.33%, while reads from intergenic regions accounted for 7.02% to 7.46% (S3 Fig). In addition, reads from the intronic areas accounted for 11.55% to 12.03% of the total. These results indicate a rational use of this study's genome and annotation files. Statistical analysis was performed on the TPM values for all gene expressions (S4 Fig ). It was observed that some outliers exist in both the OEG group and CG groups. Most gene expressions across all samples were within the range of log-transformed TPM values between 0 and 2 (i.e., TPM values range from 10 to 100).

After sequencing, the expression level of the *HADHB* gene was examined in preadipocytes infected with Ad-HADHB. The results showed a significantly higher average expression level in the OEG group compared to the CG group (Fig 4A, $P < 0.01$), which was consistent with the qPCR quantification results (Fig 4B). Differential analysis was performed. A total of 283 differentially expressed genes were identified (S4 Table), with 170 genes significantly up-regulated and 113 genes down-regulated considerably in preadipocytes of the OEG group (Fig 4C). Based on the TPM values, the top 10 expressed genes in both OEG and CG groups are listed in Table 2. In the OEG group, the three most highly expressed genes were lactate dehydrogenase A (*LDHA*), involved in amino acid metabolism, mitochondrial trifunctional protein beta-subunit (*HADHB*), involved in energy metabolism; and thrombomodulin

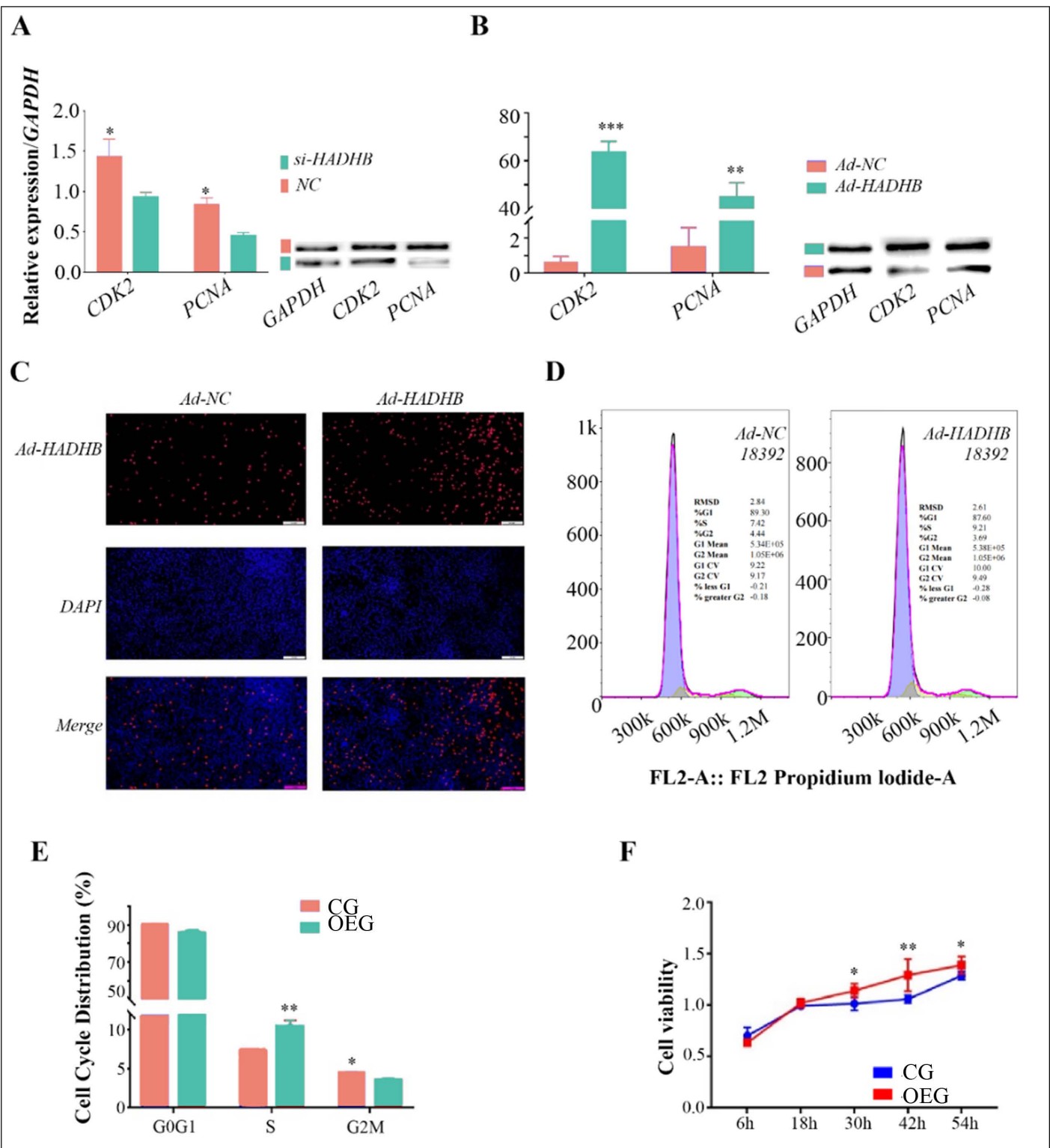

**Fig 2.** A. The effect of HADHB gene knockdown on the transcription and protein levels of the proliferation markers CDK2 and PCNA; B: The impact of HADHB gene over-expression on the transcription and protein levels of the proliferation markers CDK2 and PCNA; C: EdU assay to detect the effects of HADHB gene over-expression on the proliferation of bovine pre-adipocytes; D-E: Flow cytometry analysis to examine the impact of HADHB gene over-expression on the cell cycle of bovine pre-adipocytes; F: The effect of HADHB gene over-expression on the viability of pre-adipocytes.

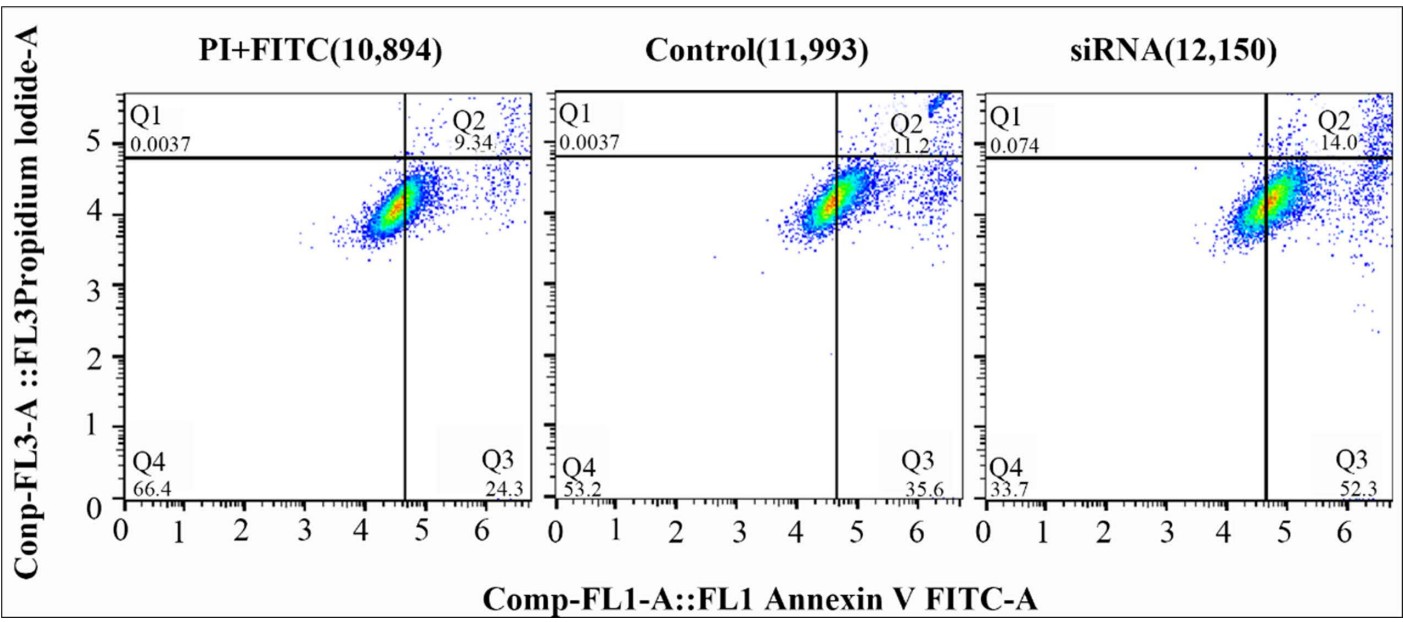

**Fig 3. Inhibition of the *HADHB* gene promotes early apoptosis of pre-adipocytes.** The horizontal and vertical axes in the figure have been transformed using a log10 scale.

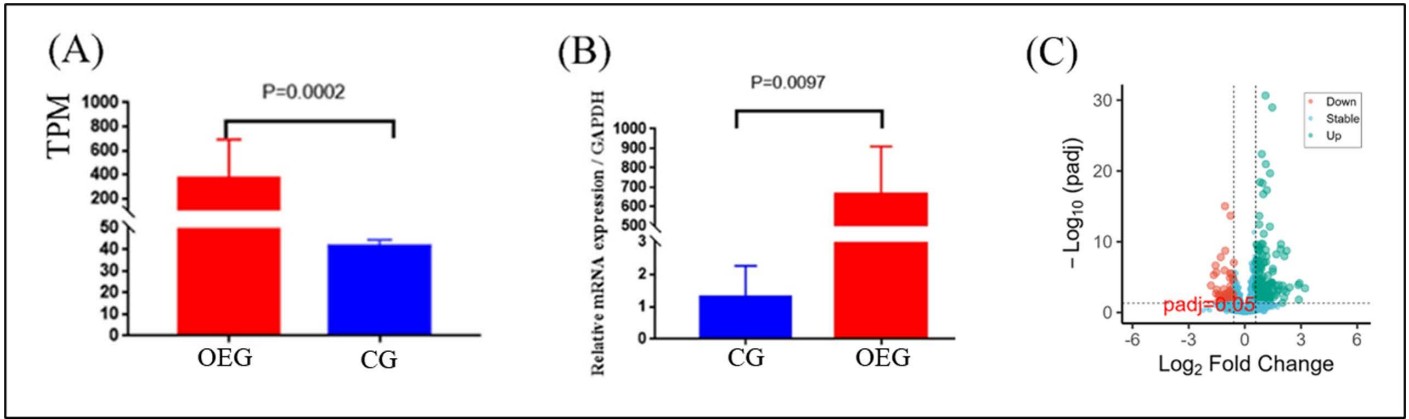

**Fig 4. Over-expression of the HADHB gene influences pre-adipocyte gene transcription levels.**

(*THBD*), involved in blood coagulation regulation. On the other hand, the three most down-regulated genes in the OEG group were chemokine (*C-X-C motif*) ligand 12 (*CXCL12*), complement component 1, subcomponent r (*C1R*) involved in inflammation, and complement component 3 (*C3*). Transcriptome analysis showed that overexpression of the *HADHB* gene resulted in significant changes in the transcriptome of preadipocytes.

### 3.5 Induction of functional changes in preadipocytes by overexpression of the *HADHB* gene in adipose tissue

Overexpression of the *HADHB* gene resulted in significant changes in the transcript levels of 283 genes, with 170 genes up-regulated and 113 genes down-regulated. Enrichment analysis of

**Table 2. Genes in the top 10 up- and down-regulated.**

| Symbol | FDR | log2FC | TPM | Regulation |
|---|---|---|---|---|
| *LDHA* | <00001 | 0.61 | 542.16 | up |
| HADHB | <00001 | 3.22 | 369.29 | up |
| THBD | <00001 | 0.78 | 249.67 | up |
| CA4 | <00001 | 0.77 | 196.59 | up |
| PTGS2 | <00001 | 0.66 | 195.61 | up |
| CDKN1A | <00001 | 0.95 | 176.87 | up |
| PHLDA3 | <00001 | 0.7 | 166.05 | up |
| PI3 | <00001 | 1.3 | 125.59 | up |
| S100A12 | <00001 | 1.18 | 125.24 | up |
| MMP1 | <00001 | 1.82 | 101.53 | up |
| SPON1 | <00001 | −0.76 | 32.86 | down |
| RRAD | 0.0001 | −1.03 | 33.13 | down |
| ECM2 | 0.0027 | −0.67 | 43.9 | down |
| SGK1 | <00001 | −0.59 | 50.83 | down |
| ACTA2 | 0.022 | −0.59 | 59.92 | down |
| COL16A1 | <00001 | −0.59 | 89.63 | down |
| SERPING1 | 0.0001 | −0.67 | 191.72 | down |
| C3 | 0.0001 | −0.66 | 232.53 | down |
| C1R | 0.0002 | −0.73 | 306.72 | down |
| CXCL12 | 0.0001 | −0.71 | 460.41 | down |

**Note**: FDR = 0 represents a P-value less than 0.00001.

the up-regulated 170 genes (Fig 5A) revealed their involvement in various signaling pathways related to lipid metabolism, such as "lipid and atherosclerosis, " "PPAR signaling pathway" and "calcium signaling pathway"; inflammation-related pathways, such as "IL-17 signaling pathway, " "inflammatory response" and "defense response"; and cell death related pathways, such as "positive regulation of cell proliferation, " "apoptosis" and "cell death"; cell death related courses, such as "positive regulation of cell death, " "positive regulation of programmed cell death" and "regulation of cell death"; and intra-peptidase activity related pathways, such as "regulation of cysteine-type endopeptidase activity, " "regulation of endopeptidase activity" and "activation of cysteine-type endopeptidase activity involved in the apoptotic process. "

In contrast, the down-regulated genes were mainly associated with cell cycle-related pathways (Fig 5B), such as "cell cycle, " "DNA replication, " "oocyte meiosis," and "DNA-templated DNA replication, " and with pathways related to the organism's immune response, such as "complement and coagulation cascades" and "complement activation. "

The functional enrichment analysis results indicated that overexpression of the *HADHB* gene enhanced signaling pathways related to lipid metabolism, inflammatory metabolism, cell cycle, and cell death in preadipocytes. This enhancement may contribute to the increased adipogenic capacity and inflammatory metabolism in preadipocytes induced by *HADHB* gene overexpression.

## 3.6 Screening of key genes in progenitor adipocytes induced by overexpression of the *HADHB* gene

Protein-protein interaction analysis revealed 172 nodes and 740 edges (Fig 6A), suggesting that the up-regulated and down-regulated genes have similar functions. Further research

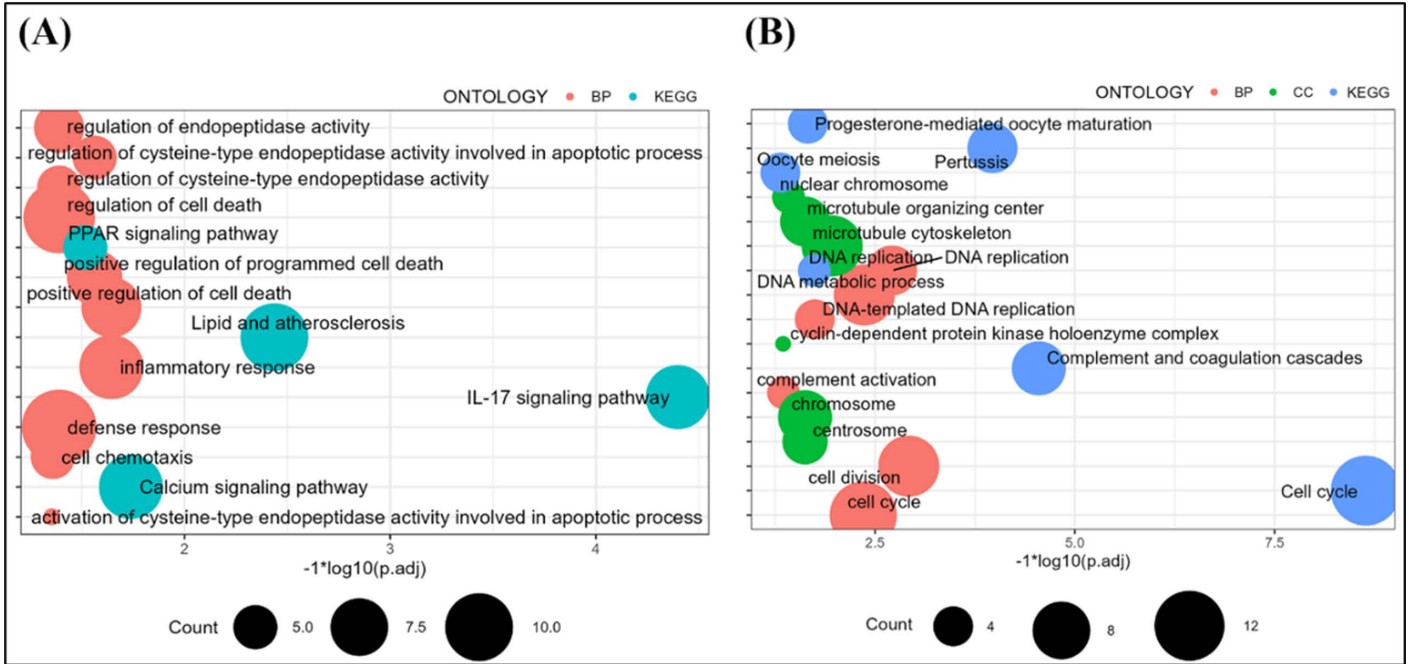

**Fig 5. Functional enrichment analysis of differentially expressed genes.** A: Functional enrichment analysis of up-regulated genes. B: Functional enrichment analysis of down-regulated genes.

using cytoHubba identified ten core genes (Fig 6, represented by a "V" shape), namely *KIF23*, *PTTG1*, *CDKN3*, *KIF4A*, *TPX2*, *MCM3*, *UHRF1*, *GINS2*, *KIF15* and *SERPING1*. Key subnetwork analysis using MCODE revealed two subnetworks (Fig.6B-C) and six seed genes (Fig 6, represented by a "diamond" shape), which were *IFI44L*, *NCAPG2*, *CDKN2B*, *KALRN*, *NGF*, and *MMP1*, with *NCAPG2* and *NGF* being seed genes for subnetwork 1 (Fig 6B) and subnetwork 2 (Fig 6C), respectively. Functional classification analysis of MCODE 1 showed that these genes (all down-regulated) were mainly involved in cell cycle-related pathways (Fig 6D), such as "cell cycle, " "DNA replication," and "FoxO signaling pathway, " among others. In MCODE 2 (Fig 6E), the up-regulated genes were predominantly involved in disease-related pathways, including the "IL-7 signaling pathway", "TNF signaling pathway, " and 'lipid and atherosclerosis. " These findings suggest that overexpression of the *HADHB* gene may alter the original functional state of preadipocytes and accelerate the cell cycle process, leading to an abnormal cellular state.

## 4. Discussion

Knockdown of the *HADHB* gene inhibits the expression of the cell proliferation markers *CDK2* and *PCNA* at both the transcriptional and translational levels. In contrast, overexpression of the *HADHB* gene has the opposite effect, increasing the proportion of cells in the S phase and decreasing the G2/M phase. Cell proliferation is regulated by a complex network of proteins that determine the progression and timing of cell cycle events. Cyclin-dependent kinase 2 (*CDK2*) plays a vital role in the G1/S transition of the cell cycle and can associate with cyclin A/D/E. The gene *CDK2* terminates the G1 to S phase transition [26,39–41]. The cell cycle cyclins (cyclins) level determines the fluctuation of CDK activity, thus regulating the cell cycle [42]. Following transcription by the transcription factor E2F, cyclin E protein accumulates and peaks during the G1/S transition of the cell cycle. After reaching its peak, cyclin E

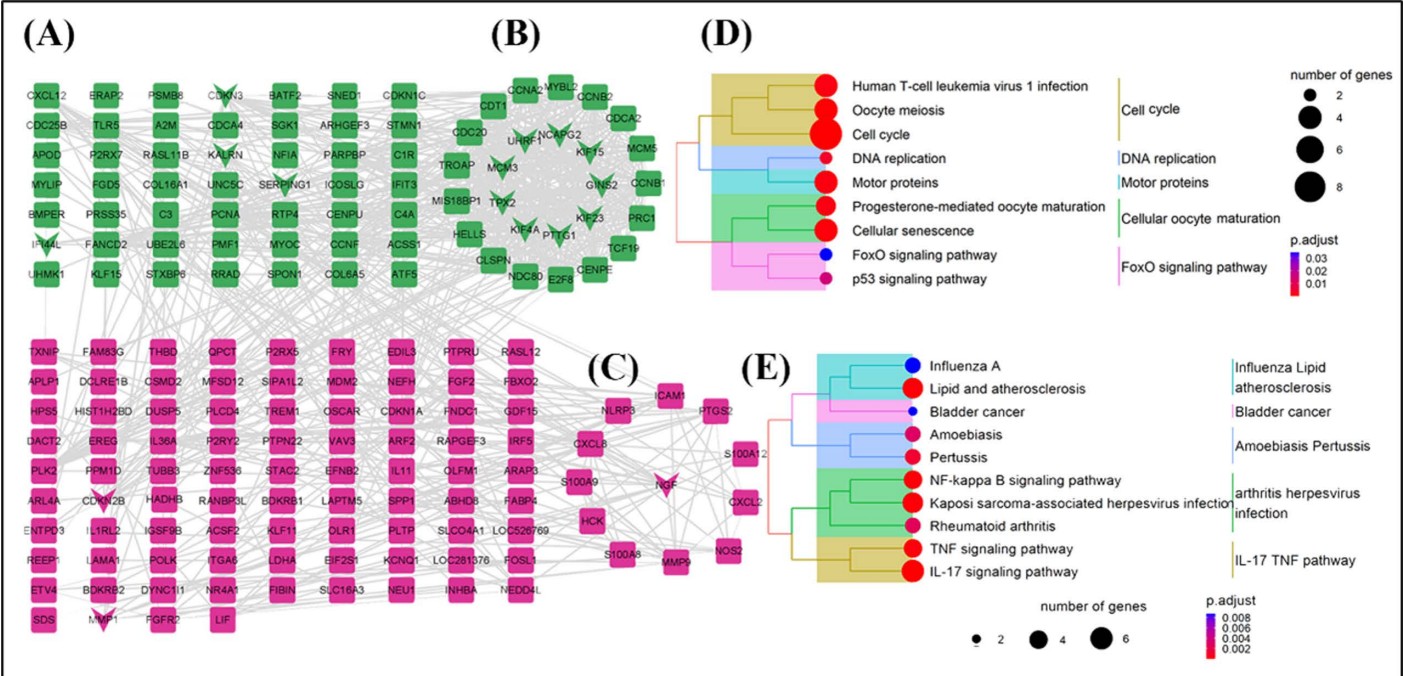

**Fig 6. Analysis of differentially expressed gene-protein interactions and functional enrichment of key sub-networks. A**-C: Protein-protein interaction analysis of differentially expressed genes; **B** and **C** represent MCODE 1 and MCODE 2 key sub-networks, respectively. **D** and **E** show genes' functional enrichment analysis results in critical sub-networks 1 and 2, respectively. In **A-C**, green represents down-regulated genes, red represents up-regulated genes, "V" represents core genes identified by the Hubba process, and the "diamond" shape represents seed genes.

binds to cyclin-dependent kinase 2 (CDK2) and promotes the transition of cells from the G1 to the S phase. The cyclin E/CDK2 complex then phosphorylates many substrates involved in cell cycle regulation, further boosting the transition from G1 to S phase. At the end of the S phase, the Cyclin E/CDK2 complex is degraded by the ubiquitin ligase complex SCFFBW7, which reduces the activity of Cyclin E/CDK2 until it is re-activated during the next cell cycle transition from G1 to S phase [43]. In addition, during the G1/S transition, cyclin E/CDK2 phosphorylates and inactivates the RB protein, thereby promoting the release of the E2F transcription factor and the transcription of cyclin E, which further promotes the G1/S transition [44]. In addition, the cyclin E/CDK2 complex could affect cell cycle progression by regulating other critical targets, such as p27KIP1 [45–47], E2F5 [48], NPAT [49], PRMT5 [50], RNF6 [51], BCR-ABL [52], thereby affecting the G1/S transition [53].

*PCNA* (proliferating cell nuclear antigen) is a gene involved in cell proliferation that forms a homotrimer and can encircle DNA and slide along the DNA to anchor DNA polymerases and other DNA repair enzymes essential for DNA replication and repair processes [54]. Studies have shown that *PCNA* is an auxiliary factor of DNA polymerases δ and ε involved in the synthesis, processing, and ligation of Okazaki fragments and recruits other elements to the DNA replication fork engaged in DNA synthesis, repair, and cell cycle control [55]. *PCNA* protein can also interact with the cyclin A/CDK2 complex during the S phase of the cell cycle, effectively stimulating the phosphorylation of CDK2 substrates, thereby targeting the DNA replication proteins anchored by *PCNA* and participating in cell cycle activities [56–59]. On the other hand, when the inhibitor p21 binds to CDK, it competitively inhibits the interaction between *PCNA*/Cyclin A/*CDK2*, leading to the termination of DNA replication and the end of the S phase, and *PCNA* has been confirmed as a critical marker gene of the cell cycle [60].

Furthermore, studies have shown that *PCNA* is mainly involved in the cell cycle's G1 to S phase transition and has its highest expression level during the S phase [61, 62].

Knockdown of the *HADHB* gene decreased transcription and translation levels of the adipogenic marker genes *PPARγ* and *CEBPα,* reduced lipid droplet formation, and induced early apoptosis in preadipocytes. Conversely, overexpression of *HADHB* had the opposite effect. Adipogenesis is a finely regulated process involving multiple transcription factors acting simultaneously during differentiation [63–66]. During the early stages of preadipocyte differentiation, transcriptional regulation is mainly controlled by CCAAT/enhancer-binding proteins (*C/EBPs*) and peroxisome proliferator-activated receptor γ (*PPARγ*) [67–71]. Under rapid induction conditions, *C/EBPβ* and *C/EBPδ* are first expressed, which in turn induce the expression of key regulators, *C/EBPα* and *PPARγ*, specific for adipocyte differentiation [72]. Ultimately, coordinated action between *C/EBPα* and *PPARγ* completes the cells' terminal differentiation, leading to mature adipocyte formation [54]. *C/EBPα*, a member of the CCAAT/enhancer-binding protein transcription factor family, is critical in promoting adipocyte differentiation. It is primarily expressed during the later stages of adipogenesis. It is one of the essential regulators of adipocyte differentiation, as evidenced by defects in white adipose tissue in mice lacking or methylating *C/EBPα* [73–76].

*PPARγ* is a PPAR nuclear receptor family member and a key adipogenesis regulator [77–79]. In addition, *PPARγ* could, in many cases, induce non-adipocyte cells to differentiate into adipocyte-like cells [80–82]. Ectopic expression of *PPARγ* in fibroblasts can drive adipogenesis, and without the presence of this gene, no other factors can induce adipogenesis [83]. Studies have also shown that *PPARγ* can induce adipogenesis in mouse embryonic fibroblasts (MEFs) lacking *C/EBPα* (C/EBPα-/-), whereas *C/EBPα* cannot induce adipogenesis in *PPARγ*-deficient (PPARγ-/-) MEFs [84]. Inhibition of *PPARγ* and *C/EBPs* significantly reduces lipid droplet formation and size [85, 86]. Once preadipocytes cease proliferation and enter the differentiation phase, they are primarily regulated by the coordinated action of *PPARγ* and *C/EBPs*, which are differentiation essential genes, ultimately leading to terminal differentiation and the formation of mature adipocytes [87–89]. Knockdown of the G0S2 gene in the mouse 3T3-L1 cell line results in decreased protein levels of *C/EBPα* and *PPARγ*, and cells begin to undergo apoptosis after 36 hours of transfection, suggesting that downregulation of *C/EBPα* and *PPARγ* expression induces apoptosis in preadipocytes [13].

## 5. Conclusions and further perspectives

The *HADHB* gene regulates the expression of *CDK2* and *PCNA*, thereby contributing to preadipocyte proliferation and maturation. *HADHB* also modulates the expression of *PPARγ* and *CEBPα*, contributing to regulating preadipocyte differentiation and lipid droplet formation. The *HADHB* gene plays a crucial role in regulating preadipocyte proliferation and maturation. In future studies, more investigations should be performed to elucidate the regulatory mechanisms of the *HADHB* gene, such as signaling pathways, transcriptional regulation, and experimental validation at the animal level.

## Supporting information

**S1 Fig. The Original gel map protein immunoblotting.**
(TIF)

**S2 Fig. PCA plot of OEG group and CG group.**
(TIF)

**S3 Fig. Statistics of reads aligned to gene structure.**
(TIF)

**S4 Fig. Box plot of TPM statistics for all genes with expression.**
(TIF)

**S1 Table. Information on Real-time quantitative PCR primer sequences.**
(XLSX)

**S2 Table. Protocols for real-time quantitative PCR.**
(XLSX)

**S3 Table. Overview of Sequencing Results.**
(XLSX)

**S4 Table. Differential analysis between OEG and CG groups.**
(XLSX)

**S1 File. S1 raw images.**
(PDF)

## Author contributions

**Conceptualization:** Ran Guan, Zengwen Huang.

**Data curation:** Shuzhe Wang, Yadong Jin.

**Formal analysis:** Chaoyun Yang, Shuzhe Wang, Yunxia Qi.

**Writing – original draft:** Chaoyun Yang.

**Writing – review & editing:** Ran Guan, Zengwen Huang.

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
