## [Decision Letter · Decision Letter 0]

7 Oct 2024

PONE-D-24-41815Mechanisms of Adipocyte Regulation: Insights from HADHB Gene ModulationPLOS ONE

Dear Dr. Huang,

Thank you for submitting your manuscript to PLOS ONE. After careful consideration, we feel that it has merit but does not fully meet PLOS ONE’s publication criteria as it currently stands. Therefore, we invite you to submit a revised version of the manuscript that addresses the points raised during the review process.

We look forward to receiving your revised manuscript.

Kind regards,

Sayed Haidar Abbas Raza

Academic Editor

PLOS ONE

Journal Requirements:

 the Ph.D. Research Launch Project of Xichang University (Grant No. YBZ202211) and Integrative technology and demonstration extension for intensive yak husbandry in Muli County (No. 2023YFN0094).  

Reviewers' comments:

Reviewer's Responses to Questions

**Comments to the Author**

1. Is the manuscript technically sound, and do the data support the conclusions?

Reviewer #1: Yes

Reviewer #2: Yes

2. Has the statistical analysis been performed appropriately and rigorously? 

Reviewer #1: Yes

Reviewer #2: Yes

3. Have the authors made all data underlying the findings in their manuscript fully available?

Reviewer #1: Yes

Reviewer #2: Yes

4. Is the manuscript presented in an intelligible fashion and written in standard English?

Reviewer #1: Yes

Reviewer #2: Yes

5. Review Comments to the Author

Reviewer #1: The manuscript entitled “Mechanisms of Adipocyte Regulation: Insights from HADHB Gene Modulation” summarized the Adipocyte proliferation and differentiation are vital for energy metabolism and lipid synthesis, ensuring energy homeostasis. The HADHB gene, associated with fatty acid metabolism, was studied for its role in regulating bovine preadipocyte proliferation and differentiation. Findings revealed that HADHB modulates key genes influencing these processes and alters cell cycle progression, providing insights into adipocyte biology regulation. Generally, the manuscript is well written. This paper has several weaknesses and needs improvement before publication.

This manuscript has major language problems. There are too many for me to modify them all. Authors are strongly encouraged to seek a native English speaker who may assist you modifying the document.

Comments:

1. Please write your full affiliation with your college you only written the university name ?

2. What cell type was utilized for RNA extraction

3. How much RNA was used in PCR? Was it one-step or 2-step PCR?

4. The abstract is not particularly informative and would benefit from more background.

5. Summarize the abstract, focus on the main findings and mention the small conclusion in at the end of abstract

6. In the Introduction focus on the objectives and insert a few new reference and relevant findings

7. In material and method sections, references are missing.

8. Most of the references mentioned are old and I suggest adding recent references, and the manuscript should be edited accordingly.

9. I suggest the cite following paper in introduction part For more information you can read below reference

Investigating the role of KLF6 in the growth of bovine preadipocytes: using transcriptomic analyses to understand beef quality. Journal of Agricultural and Food Chemistry, 72(17), pp.9656-9668.Material and method needs to clarifying and summarizing- some detailed needs

The subtitles in the material and method needs to summarizing Ethical approval and references must be mentioned in M&M

Reviewer #2: The purpose of the manuscript "Mechanisms of Adipocyte Regulation: Insights from HADHB Gene Modulation" was to learn more about the mechanisms behind adipocyte regulation while examining the impact of the HADHB gene on the differentiation and proliferation of bovine preadipocytes. The authors performed both knock down and overexpression experiments. Based on the results, it appears that the HADHB gene regulates the expression of adipogenic marker genes CEBPα and PPARγ, which may play a role in adipocyte differentiation. Additionally, the transcriptome study revealed that the transcriptional profile of preadipocytes was considerably altered by HADHB gene overexpression. The HADHB gene, according to the authors, controls PCNA and CDK2 expression, which promotes preadipocyte maturation and proliferation. In addition, HADHB controls the production of CEBPα and PPARγ, which helps to control preadipocyte development and the creation of lipid droplets. This manuscript is very well written. The methods are described in detail and the conclusions are based on the results. Figures are well prepared and well labelled. I congratulate the authors for performing a thorough study to decipher the role of HDAHB in adipocyte regulation.

6. PLOS authors have the option to publish the peer review history of their article (what does this mean? ). If published, this will include your full peer review and any attached files.

**Do you want your identity to be public for this peer review?** For information about this choice, including consent withdrawal, please see our Privacy Policy .

Reviewer #1: No

Reviewer #2: No

---

## [Author Response · Author response to Decision Letter 1]

11 Dec 2024

PONE-D-24-41815

Mechanisms of Adipocyte Regulation: Insights from HADHB Gene Modulation

Dear Editor,

Please accept our sincere apologies for the delay in submitting the revised manuscript. We would like to express our gratitude to the editor and reviewers for their insightful comments on the article, which have significantly enhanced its quality. The revisions to the article are summarized as follows:

First, the manuscript and images were meticulously examined to ensure compliance with the journal's requirements. Second, the original WB images were provided in Fig. S1. Finally, the article was revised in accordance with the reviewers' comments and updated with a substantial number of references. Additionally, the funding statement was revised to read, "The funders had no role in study design, data collection and analysis, decision to publish, or preparation of the manuscript."

Reviewer #1: The manuscript entitled “Mechanisms of Adipocyte Regulation: Insights from HADHB Gene Modulation” summarized the Adipocyte proliferation and differentiation are vital for energy metabolism and lipid synthesis, ensuring energy homeostasis. The HADHB gene, associated with fatty acid metabolism, was studied for its role in regulating bovine preadipocyte proliferation and differentiation. Findings revealed that HADHB modulates key genes influencing these processes and alters cell cycle progression, providing insights into adipocyte biology regulation. Generally, the manuscript is well written. This paper has several weaknesses and needs improvement before publication.

This manuscript has major language problems. There are too many for me to modify them all. Authors are strongly encouraged to seek a native English speaker who may assist you modifying the document.

Comments:

1. Please write your full affiliation with your college you only written the university name ?

Response: The author's institutional affiliation has been updated to reflect their affiliation with the College of Animal Science and Technology at Xichang University, located in Xichang, China. Please refer to the details provided on line 4.

2. What cell type was utilized for RNA extraction。

Response: The precursor cells of adipose tissue. In Part 2.1, adipose tissue was utilized for RNA extraction with the objective of obtaining the CDS sequence of the HADHB gene. In Part 2.6, the preadipocytes were utilized for the purpose of RNA extraction.

3.How much RNA was used in PCR? Was it one-step or 2-step PCR?

Response: A one-step PCR amplification of the HADHB gene CDS region was conducted using 2 ng of cDNA (see Supplemental L87-L88 for details).

4.The abstract is not particularly informative and would benefit from more background.

Response: In order to provide a more comprehensive account, we have augmented the Abstract part with a detailed description of the methodologies employed and the findings attained. We therefore request that you peruse the revised manuscript.

5.Summarize the abstract, focus on the main findings and mention the small conclusion in at the end of abstract

Response: We have provided a summary of the Abstract and a conclusion. Please review the revised manuscript.

6.In the Introduction focus on the objectives and insert a few new reference and relevant findings

Response: The references in the summary and discussion sections have been updated and some newer literature has been incorporated. Please review the revised manuscript.

7.In material and method sections, references are missing.

Response: A moderate amount of literature on the use of software has been included in the Material Methods section. Please review the revised manuscript.

8.Most of the references mentioned are old and I suggest adding recent references, and the manuscript should be edited accordingly.

Response: The document has been updated and new references from recent years have been added. Please refer to the revised manuscript.

9.I suggest the cite following paper in introduction part For more information you can read below reference：Investigating the role of KLF6 in the growth of bovine preadipocytes: using transcriptomic analyses to understand beef quality. Journal of Agricultural and Food Chemistry, 72(17), pp.9656-9668.

Response: In the Introduction, we cite this literature (reference 14) in order to enrich the list of fat-regulating factors.。

10.Material and method needs to clarifying and summarizing- some detailed needs

The subtitles in the material and method needs to summarizing Ethical approval and references must be mentioned in M&M

Response: In accordance with the latest developments in ethical standards, an additional statement has been incorporated into section 2.1 of the revised draft, which is available for your perusal.

---

## [Decision Letter · Decision Letter 1]

2 Jan 2025

PONE-D-24-41815R1Mechanisms of Adipocyte Regulation: Insights from HADHB Gene ModulationPLOS ONE

Dear Dr. Huang,

Thank you for submitting your manuscript to PLOS ONE. After careful consideration, we feel that it has merit but does not fully meet PLOS ONE’s publication criteria as it currently stands. Therefore, we invite you to submit a revised version of the manuscript that addresses the points raised during the review process.

We look forward to receiving your revised manuscript.

Kind regards,

Sayed Haidar Abbas Raza

Academic Editor

PLOS ONE

Journal Requirements:

Reviewers' comments:

Reviewer's Responses to Questions

**Comments to the Author**

1. If the authors have adequately addressed your comments raised in a previous round of review and you feel that this manuscript is now acceptable for publication, you may indicate that here to bypass the “Comments to the Author” section, enter your conflict of interest statement in the “Confidential to Editor” section, and submit your "Accept" recommendation.

Reviewer #1: All comments have been addressed

Reviewer #2: (No Response)

2. Is the manuscript technically sound, and do the data support the conclusions?

Reviewer #1: Yes

Reviewer #2: Yes

3. Has the statistical analysis been performed appropriately and rigorously? 

Reviewer #1: Yes

Reviewer #2: Yes

4. Have the authors made all data underlying the findings in their manuscript fully available?

Reviewer #1: Yes

Reviewer #2: Yes

5. Is the manuscript presented in an intelligible fashion and written in standard English?

Reviewer #1: Yes

Reviewer #2: Yes

6. Review Comments to the Author

Reviewer #1: The authors should be commended for presenting a much improved article. I am happy for the article to be published

Reviewer #2: This manuscript by Yang et al focuses on the role of the HADHB gene, which encodes the beta-subunit of 3-hydroxy acyl-CoA dehydrogenase, in regulating bovine preadipocyte proliferation and differentiation. The authors have shown that HADHB plays an important role in the regulation of adipocyte proliferation and differentiation, which is a critical process in energy metabolism and lipid synthesis. By applying techniques like siRNA-mediated knockdown, adenoviral overexpression, transcriptomic sequencing, and functional enrichment analysis, the authors identified PPARγ and CEBPα in regulating differentiation. CDK2 and PCNA, were identified as critical for proliferation. Knockdown of HADHB inhibited differentiation, reduced lipid droplet formation, while its overexpression enhanced the processes. Furthermore, knockdown of HADHB promoted early-stage apoptosis in preadipocytes, suggesting its critical role in cell survival. By transcriptomic analysis, the authors pointed out that overexpression of HADHB induced changes in the expression of 283 genes and enrichment of pathways related to lipid metabolism, cell cycle regulation, inflammation, and apoptosis. Protein-protein interaction analysis revealed key genes such as KIF23, SERPING1, and others that are crucial in the regulation of these processes. The manuscript identifies that HADHB modulates the preadipocyte biology by influencing transcriptional profiles and functional pathways, thus its putative role in energy homeostasis and metabolic regulation. The authors propose that further studies should be conducted to determine the signaling pathways and mechanisms through which HADHB mediates such effects, laying a foundation for exploring its role in the regulation of fat traits in livestock and human metabolic disorders. The manuscript is well written. I have a few suggestions to improve the manuscript

1. The authors need to address the limitations of the study.

2. The manuscript does not adequately emphasize the broader implications of the findings. The authors can discuss how the findings could inform strategies for improving livestock productivity.

7. PLOS authors have the option to publish the peer review history of their article (what does this mean? ). If published, this will include your full peer review and any attached files.

**Do you want your identity to be public for this peer review?** For information about this choice, including consent withdrawal, please see our Privacy Policy .

Reviewer #1: No

Reviewer #2: No

---

## [Author Response · Author response to Decision Letter 2]

29 Jan 2025

PONE-D-24-41815

Mechanisms of Adipocyte Regulation: Insights from HADHB Gene Modulation

Dear Editor,

Please accept our sincere apologies for the delay in submitting the revised manuscript. We would like to express our gratitude to the editor and reviewers for their insightful comments on the article, which have significantly enhanced its quality. The revisions to the article are summarized as follows:

First, the manuscript and images were meticulously examined to ensure compliance with the journal's requirements. Second, the original WB images were provided in Fig. S1. Finally, the article was revised in accordance with the reviewers' comments and updated with a substantial number of references. Additionally, the funding statement was revised to read, "The funders had no role in study design, data collection and analysis, decision to publish, or preparation of the manuscript."

Reviewer #1: The manuscript entitled “Mechanisms of Adipocyte Regulation: Insights from HADHB Gene Modulation” summarized the Adipocyte proliferation and differentiation are vital for energy metabolism and lipid synthesis, ensuring energy homeostasis. The HADHB gene, associated with fatty acid metabolism, was studied for its role in regulating bovine preadipocyte proliferation and differentiation. Findings revealed that HADHB modulates key genes influencing these processes and alters cell cycle progression, providing insights into adipocyte biology regulation. Generally, the manuscript is well written. This paper has several weaknesses and needs improvement before publication.

This manuscript has major language problems. There are too many for me to modify them all. Authors are strongly encouraged to seek a native English speaker who may assist you modifying the document.

Comments:

1. Please write your full affiliation with your college you only written the university name ?

Response: The author's institutional affiliation has been updated to reflect their affiliation with the College of Animal Science and Technology at Xichang University, located in Xichang, China. Please refer to the details provided on line 4.

2. What cell type was utilized for RNA extraction。

Response: The precursor cells of adipose tissue. In Part 2.1, adipose tissue was utilized for RNA extraction with the objective of obtaining the CDS sequence of the HADHB gene. In Part 2.6, the preadipocytes were utilized for the purpose of RNA extraction.

3.How much RNA was used in PCR? Was it one-step or 2-step PCR?

Response: A one-step PCR amplification of the HADHB gene CDS region was conducted using 2 ng of cDNA (see Supplemental L87-L88 for details).

4.The abstract is not particularly informative and would benefit from more background.

Response: In order to provide a more comprehensive account, we have augmented the Abstract part with a detailed description of the methodologies employed and the findings attained. We therefore request that you peruse the revised manuscript.

5.Summarize the abstract, focus on the main findings and mention the small conclusion in at the end of abstract

Response: We have provided a summary of the Abstract and a conclusion. Please review the revised manuscript.

6.In the Introduction focus on the objectives and insert a few new reference and relevant findings

Response: The references in the summary and discussion sections have been updated and some newer literature has been incorporated. Please review the revised manuscript.

7.In material and method sections, references are missing.

Response: A moderate amount of literature on the use of software has been included in the Material Methods section. Please review the revised manuscript.

8.Most of the references mentioned are old and I suggest adding recent references, and the manuscript should be edited accordingly.

Response: The document has been updated and new references from recent years have been added. Please refer to the revised manuscript.

9.I suggest the cite following paper in introduction part For more information you can read below reference：Investigating the role of KLF6 in the growth of bovine preadipocytes: using transcriptomic analyses to understand beef quality. Journal of Agricultural and Food Chemistry, 72(17), pp.9656-9668.

Response: In the Introduction, we cite this literature (reference 14) in order to enrich the list of fat-regulating factors.。

10.Material and method needs to clarifying and summarizing- some detailed needs

The subtitles in the material and method needs to summarizing Ethical approval and references must be mentioned in M&M

Response: In accordance with the latest developments in ethical standards, an additional statement has been incorporated into section 2.1 of the revised draft, which is available for your perusal.

---

## [Decision Letter · Decision Letter 2]

2 Feb 2025

Mechanisms of Adipocyte Regulation: Insights from HADHB Gene Modulation

PONE-D-24-41815R2

Dear Dr. Huang,

We’re pleased to inform you that your manuscript has been judged scientifically suitable for publication and will be formally accepted for publication once it meets all outstanding technical requirements.

Kind regards,

Sayed Haidar Abbas Raza

Academic Editor

PLOS ONE

Additional Editor Comments (optional):

Reviewers' comments:

Reviewer's Responses to Questions

**Comments to the Author**

1. If the authors have adequately addressed your comments raised in a previous round of review and you feel that this manuscript is now acceptable for publication, you may indicate that here to bypass the “Comments to the Author” section, enter your conflict of interest statement in the “Confidential to Editor” section, and submit your "Accept" recommendation.

Reviewer #1: (No Response)

Reviewer #2: All comments have been addressed

2. Is the manuscript technically sound, and do the data support the conclusions?

Reviewer #1: Yes

Reviewer #2: Yes

3. Has the statistical analysis been performed appropriately and rigorously? 

Reviewer #1: Yes

Reviewer #2: Yes

4. Have the authors made all data underlying the findings in their manuscript fully available?

Reviewer #1: Yes

Reviewer #2: Yes

5. Is the manuscript presented in an intelligible fashion and written in standard English?

Reviewer #1: Yes

Reviewer #2: Yes

6. Review Comments to the Author

Reviewer #1: The revised manuscript now looks much improved and the authors have tried to respond to most of my queries raised during the initial review. I have no other queries.

Reviewer #2: The authors have addressed all the concerns. The revised version of the manuscript has taken into account my suggestions.

7. PLOS authors have the option to publish the peer review history of their article (what does this mean? ). If published, this will include your full peer review and any attached files.

**Do you want your identity to be public for this peer review?** For information about this choice, including consent withdrawal, please see our Privacy Policy .

Reviewer #1: No

Reviewer #2: No

---

## [Editor Report · Acceptance letter]

PONE-D-24-41815R2

PLOS ONE

Dear Dr. Huang,

I'm pleased to inform you that your manuscript has been deemed suitable for publication in PLOS ONE. Congratulations! Your manuscript is now being handed over to our production team.

Kind regards,

on behalf of

Dr. Sayed Haidar Abbas Raza

Academic Editor

PLOS ONE